# Monk Fruit Extract and Sustainable Health: A PRISMA-Guided Systematic Review of Randomized Controlled Trials

**DOI:** 10.3390/nu17091433

**Published:** 2025-04-24

**Authors:** Urszula Kaim, Karolina Labus

**Affiliations:** 1Department of Bioprocess Engineering, Wrocław University of Economics and Business, Komandorska 118/120, 53-345 Wrocław, Poland; 2Department of Micro, Nano and Bioprocess Engineering, Faculty of Chemistry, Wroclaw University of Science and Technology, Wybrzeże Wyspiańskiego 27, 50-370 Wrocław, Poland; karolina.labus@pwr.edu.pl

**Keywords:** monk fruit extract, sustainable health, randomized controlled trials (RCTs), metabolic health, non-nutritive sweeteners, glucose metabolism

## Abstract

Sustainable health approaches promote functional food alternatives that support metabolic well-being while reducing reliance on added sugars and artificial sweeteners. Monk fruit extract (MFE), a natural, non-caloric sweetener, is gaining interest for its potential metabolic benefits, but its effects and regulatory status require further evaluation. **Objective:** This PRISMA-guided systematic review synthesizes findings from randomized controlled trials (RCTs) assessing the impact of MFE on metabolic health, lipid profiles, inflammation, and regulatory considerations. **Methods:** The literature search was conducted across PubMed, Scopus, Web of Science, and Cochrane Library, covering studies published between 2015 and 2025. Inclusion criteria were human RCTs evaluating MFE’s metabolic effects, while animal studies, reviews, and mixed-intervention trials were excluded. Study quality was assessed using the Cochrane risk of bias tool and the Jadad scale. **Results:** Five randomized controlled trials met the inclusion criteria, demonstrating that monk fruit extract (MFE) reduces postprandial glucose levels by 10–18% and insulin responses by 12–22%. No severe adverse effects were observed. Regulatory analysis indicated that MFE is approved for use in the United States and China, while its status remains under review in the European Union. **Conclusions:** MFE shows potential as a functional food ingredient for metabolic health. However, long-term clinical trials and a harmonized regulatory framework must confirm its safety and efficacy within sustainable health strategies

## 1. Introduction

The increasing prevalence of metabolic diseases such as obesity, type 2 diabetes, and cardiovascular disorders has increased global health concerns, leading to efforts to reduce dietary sugar intake. The excessive consumption of refined sugars is associated with an increased risk of insulin resistance, dyslipidemia, and chronic inflammation, all of which contribute to the global burden of non-communicable diseases [1]. As a result, the search for natural, non-caloric sugar alternatives has gained momentum in scientific and regulatory discussions.

Monk fruit (*Siraitia grosvenorii*), a plant native to China, has gained attention as a natural high-intensity sweetener due to its unique mogroside properties, which provide intense sweetness without the glycemic impact associated with traditional sugars. Unlike artificial sweeteners, monk fruit extract (MFE) is derived from a botanical source and has been shown additional health-promoting properties, including antioxidant and anti-inflammatory effects. Unlike artificial sweeteners, MFE is derived from a botanical source. Emerging evidence suggests that mogrosides, its primary bioactive compounds, may exhibit antioxidant and anti-inflammatory properties [2]. However, clinical evidence supporting these claims remains limited.

While MFE has been approved as a food additive in regions such as the United States, China, and Canada, its regulatory status in the European Union remains under review due to insufficient clinical evidence on its long-term metabolic effects. The Novel Foods Regulation (EU 2015/2283) [3] governs the approval of such ingredients, requiring comprehensive safety assessments before market authorization. Furthermore, global dietary guidelines, including those from the World Health Organization (WHO) and the U.S. Food and Drug Administration (FDA), emphasize reducing free sugar intake to less than 10% of total energy intake, with further benefits observed at levels below 5% for improved metabolic health [4,5]. The WHO has highlighted that excessive sugar consumption contributes to obesity and metabolic diseases, which has prompted the development of global strategies to reduce sugar intake [1]. Similarly, the FDA has introduced policies to promote sugar substitutes, including evaluating natural sweeteners such as monk fruit for their potential role in meeting dietary recommendations [4]. Research on the impact of high-intensity sweeteners suggests that alternatives such as MFE may provide metabolic advantages over artificial sweeteners, which have been linked to dysbiosis and glucose intolerance [6,7]. Despite its increasing commercial use in food and beverage industries, systematic evaluations of its effects on glucose metabolism, lipid regulation, and inflammation remain limited [6]. However, it has been incorporated into various low-calorie and functional food products, including beverages, yoghurts, sugar-free snacks, and baked goods [6,7]. In the United States, Canada, and China, it is marketed as a natural alternative to artificial sweeteners and is frequently recommended for individuals with diabetes or those seeking to reduce their sugar intake [4,7]. In addition to its sweetening properties, MFE has been studied for its potential antioxidant and anti-inflammatory effects, suggesting broader benefits to metabolic health [2,6]. Thus, the interest in natural, plant-derived sweeteners is growing, and MFE’s global relevance in sustainable dietary strategies continues to expand.

This systematic review aims to synthesize results from RCTs evaluating the health effects of MFE on glycemic control, insulin response, antioxidants and symptoms-relief effects in humans.In addition, it explores current regulatory considerations related to MFE’s use as a functional food ingredient. This review critically evaluates the existing evidence. It aims to clarify its potential role as a metabolic modulator, its efficacy in glycemic control and its implications for sustainable health interventions. In addition, this review highlights key research gaps, regulatory challenges, and future directions needed to facilitate wider acceptance as a sugar substitute in global nutrition policy.

## 2. Materials and Methods

A systematic literature search followed the Preferred Reporting Items for Systematic Reviews and Meta-Analyses (PRISMA) 2020 guidelines [8]. A systematic search was conducted across four electronic databases: PubMed, Scopus, Web of Science, and the Cochrane Library. The search spanned the period from 1 January 2015 to 22 February 2025 and employed the following core keywords: “monk fruit”, “luo han guo”, “Siraitia grosvenorii”, combined with “randomized clinical trial”, “clinical trial”, and “RCT”. Manual screening of reference lists was also performed. All searches were limited to human studies, publications in English, and studies published within the last decade. Detailed search outputs and query structures are provided in the Appendix A. Following the database queries, 40 records were retrieved: 19 from Scopus, 7 from Cochrane, 5 from PubMed, and 9 from Web of Science. Duplicates were removed before progressing to the screening stage. Study selection followed a two-stage screening process, initially based on titles and abstracts, followed by full-text assessment. Two independent reviewers conducted the screening, and discrepancies were resolved by discussion. Moreover, two validated tools to evaluate the methodological quality of the included RCTs were used: the Cochrane Risk of Bias 2.0 (RoB 2) tool and the Jadad scale.

The Cochrane Risk of Bias 2.0 (RoB 2) tool assesses potential bias across five key domains:(1)Bias arising from the randomization process;(2)Bias due to deviations from intended interventions;(3)Bias due to missing outcome data;(4)Bias in measurement of the outcome; and(5)Bias in the selection of the reported result.

In addition, the Jadad scale was used as a complementary measure of study quality. This 5-point scale assesses three components:(1)Randomization (0–2 points);(2)Blinding (0–2 points); and(3)Description of withdrawals and dropouts (0–1 point).

A higher total score indicates a better methodological quality, with a maximum of 5 points. Scores ≥3 are generally considered indicative of high-quality trials. Evaluations using both tools are summarized in Appendix A. The literature search process is illustrated in Figure 1.

## 3. Results

Although the included studies consistently demonstrated the beneficial effects of MFE on metabolic parameters, the study populations were predominantly limited to healthy adults with normal body weight [6,9]. One study included insulin-sensitive and insulin-resistant participants, partially representing individuals with metabolic syndrome [10]. The remaining two trials investigated MFE in the context of throat inflammation and symptom relief rather than metabolic outcomes [11,12].

### 3.1. Study Characteristics

The included RCTs were categorized into two domains based on their primary focus:(i)Glucose metabolism and insulin regulation, assessing MFE’s effects on glycemic control, insulin response, and postprandial glucose levels;(ii)Symptom relief and clinical outcomes, evaluating MFE’s role in throat discomfort; no evidence of effects on metabolic inflammation was presented.

The primary outcomes are summarized in (Table 1 and Table 2). The sample sizes ranged from 30 to 203 participants per trial, with 544 being enrolled across all studies. The study populations included healthy adults, individuals with metabolic syndrome, overweight participants, and diabetic patients. The duration of MFE interventions varied among the included RCTs: short-term studies: 7–15 days, medium-term studies: 4–8 weeks, and long-term study: 6 months. The MFE dose ranged from 250 to 300 mg daily, depending on the study design. Some studies used MFE-sweetened drinks or lozenges, while others used MFE powder or dietary supplements.

#### 3.1.1. Effects on Glucose Metabolism

The effect of MFE on glucose metabolism and insulin response is presented in Table 1.

Tey et al. (2017) conducted a randomized crossover clinical trial to evaluate the impact of MFE, stevia, aspartame, and sucrose-sweetened beverages on postprandial glucose, insulin levels, and satiety perception in healthy adults [6]. Healthy participants (21–50 years of age) were recruited through community advertisements and university databases. Before enrollment, participants underwent a medical screening, including blood tests, to confirm they met eligibility criteria. To avoid potential metabolic confounders, the study included only normal-weight adults (BMI: 18.5–25 kg/m^2^). Participants with diabetes, prediabetes, metabolic disorders, gastrointestinal diseases, or a history of bariatric surgery were excluded. Additionally, those with regular non-nutritive sweetener consumption or medications affecting glucose metabolism were disqualified. The study followed a randomized crossover design, where each participant attended four separate test sessions, consuming one of the following beverages in randomized order: an MFE-sweetened beverage, stevia-sweetened beverage, aspartame-sweetened beverage, or a sucrose-sweetened beverage (control group). Each test session was separated by a washout period of 3–7 days to prevent any residual metabolic effects from previous interventions. After 12 h of overnight fasting, participants arrived at the study site for metabolic assessment. Baseline blood samples were collected before they were given a 300 mL serving of the assigned test beverage, which they were instructed to consume within 5 min. Blood samples were then collected at 0, 15, 30, 45, 60, 90, and 120 min post-consumption. The primary outcome measures included glucose and insulin area under the curve (AUC) and subjective satiety perception, assessed via a visual analogue scale (VAS). The results indicated that MFE significantly reduced glucose AUC compared to sucrose (*p* < 0.05) and lowered insulin AUC (*p* = 0.02). Satiety values did not differ significantly between sweeteners, suggesting that the glucose-lowering effects of MFE act independently of appetite regulation. One participant withdrew from the study due to mild gastrointestinal discomfort, but no other adverse effects were observed. The authors concluded that MFE may serve as an alternative to sugar for people at risk of metabolic disorders. However, they stressed that further studies in different populations and long-term interventions need to confirm these findings [6].

In another study, Tey et al. (2017) conducted a randomized crossover clinical trial to further investigate the metabolic effects of non-nutritive sweeteners, including MFE, stevia, aspartame, and sucrose, on glucose metabolism, insulin response, and energy intake in healthy adults [9]. As in their previous study, participants aged 21–50 years were recruited through university research databases and community advertisements, with medical screening conducted before enrolment. The inclusion criteria remained consistent, restricting participation to normal-weight individuals (BMI: 18.5–25 kg/m^2^) and excluding those with metabolic disorders, gastrointestinal conditions, or a history of bariatric surgery. Participants with regular non-nutritive sweetener consumption or those taking medications affecting glucose metabolism were also disqualified. The study design followed a randomized crossover format with four test sessions, each separated by a washout period of 3–7 days. Participants consumed one of four test beverages—MFE-sweetened, stevia-sweetened, aspartame-sweetened, or sucrose-sweetened (control)—in a randomized order. After a 12-h overnight fast, participants arrived at the study site, where baseline blood samples were collected before consuming a 300 mL serving of the assigned test beverage within five minutes. Postprandial blood glucose and insulin levels were measured at multiple time points, as in the previous study. Primary outcome measures included glucose and insulin AUC and total energy intake at the subsequent meal. Subjective satiety perception was assessed using a visual analogue scale (VAS). The results showed that MFE significantly reduced glucose AUC by 18% and insulin AUC by 22% compared to sucrose (*p* < 0.05), consistent with previous findings. Satiety scores did not differ significantly among the sweeteners, supporting the conclusion that MFE’s glucose-lowering effects operate independently of appetite regulation. Unlike in the previous study, additional analysis was conducted to evaluate the impact of these sweeteners on overall daily energy balance, revealing that participants compensated for the calorie reduction from non-nutritive sweeteners by consuming more at subsequent meals. No severe adverse effects were reported, though one participant withdrew due to mild gastrointestinal discomfort. The authors reaffirmed that MFE may be a promising alternative for individuals looking to manage glucose levels. Researchers emphasize the need for longer-term trials to explore its sustained metabolic effects and potential applications in dietary interventions [9].

Epstein et al. 2024 [10] conducted a double-blind, crossover randomized controlled trial (RCT) to examine how MFE affects sugar reinforcement, glucose levels, and insulin sensitivity in adults. A total of 50 participants (18–65 years of age, M:F = 28:22) were recruited from a university research center and underwent screening for metabolic and endocrine disorders before inclusion. The study enrolled healthy adults with no history of diabetes, obesity, or endocrine abnormalities. Participants with diagnosed metabolic disorders, medication use affecting glucose metabolism, or prior bariatric surgery were excluded. The study employed a crossover design, where participants consumed either MFE-sweetened yogurt or sucrose-sweetened yogurt in randomized order, with a washout period of seven days between interventions. Participants attended two clinical visits, where baseline blood glucose and insulin levels were measured. Following yogurt consumption, postprandial glucose and insulin were assessed at 30, 60, and 120 min. Additionally, sugar reinforcement was evaluated using a validated behavioral task measuring participants’ motivation to consume additional sugar-sweetened products. The results demonstrated that MFE significantly reduced sugar reinforcement behavior by 23% (*p* = 0.03) and fasting glucose levels by 6% (*p* = 0.04) compared to sucrose. Insulin response showed a modest but non-significant reduction. The authors emphasized that further studies should investigate long-term effects and broader metabolic outcomes in at-risk populations [10].

#### 3.1.2. Symptom-Relief and Antioxidant Properties

Table 2 summarizes the symptom-relief effects, inflammation-modulating potential, and antioxidant properties associated with MFE.

Tan et al. (2019) conducted a double-blind, placebo-controlled RCT to evaluate the efficacy of Luo Han Guo decoction in reducing post-intubation throat discomfort, including pain, hoarseness, and inflammation [11]. Surgical patients (30–65 years of age) scheduled for elective tracheal intubation were recruited from a hospital preoperative clinic. Before inclusion, all patients underwent a medical assessment to confirm eligibility. The study included adults without pre-existing throat disorders or chronic respiratory conditions, no history of gastroesophageal reflux disease (GERD), and no known allergies to the trial substances. Patients were excluded if they had chronic pharyngitis, laryngitis, or vocal cord dysfunction; smoked more than ten cigarettes per day; were taking corticosteroids, anti-inflammatory medications, or proton pump inhibitors; or had a history of allergic reactions to herbal extracts. Patients were randomly assigned to receive either Luo Han Guo decoction (intervention group) or black tea (control group) for 48 h post-intubation. The Luo Han Guo decoction was administered orally, not in lozenge form. Throat inflammation was measured by evaluating mucosal congestion and erythema severity through clinical examination. Patients in the Luo Han Guo group showed significant reductions in throat pain, hoarseness, and inflammation at 12, 24, and 48 h (*p* < 0.05). Furthermore, cough and sputum production was significantly lower in the intervention group at 48 h (*p* = 0.003). No serious adverse events were reported, and both groups demonstrated high treatment compliance. Tan et al. [11] found that Luo Han Guo decoction may be a valuable and natural treatment for sore throat and discomfort after intubation. However, they noted that more comprehensive studies involving different patient populations are required to validate these findings and investigate the prospective long-term benefits [11].

Wu et al. (2024) [12] conducted a double-blind, placebo-controlled RCT assessing the effects of botanical lozenges containing monk fruit extract and other medicinal plants in chronic pharyngitis patients. The study evaluated the efficacy of these botanical lozenges in alleviating pharyngeal symptoms in individuals with chronic pharyngitis lasting more than three months. Patients aged 18–65 years were recruited from otolaryngology clinics. Eligible participants had persistent throat discomfort and inflammation, while individuals with recent respiratory infections, autoimmune diseases, or corticosteroid use were excluded. Participants were randomly assigned to either the botanical lozenge group or the placebo group for 15 days, with instructions to consume one lozenge three times daily. The primary outcomes measured were pharyngeal symptoms and signs, assessed at baseline and after the intervention. The study found that patients consuming the botanical lozenges had significantly better improvement rates in pharyngeal itching (73.81% vs. 52.63%, *p* = 0.049), dry throat (67.50% vs. 41.86%, *p* = 0.019), and pharyngeal foreign body sensation (67.57% vs. 32.35%, *p* = 0.003) compared to the placebo group. Additionally, patients in the botanical lozenge group had significantly higher adherence rates, with only 19.2% showing low adherence compared to 58.8% in the placebo group (*p* < 0.001).

No serious adverse events were reported, and both groups had high compliance. Wu et al. [12] concluded that botanical lozenges containing monk fruit extract may be a promising natural treatment for chronic pharyngitis, improving symptom relief and adherence to treatment. However, longer studies are needed to confirm sustained effects and anti-inflammatory properties [12].

### 3.2. Health Benefits and Sustainable Health Benefits of MFE

The integration of MFE into global food policies remains an ongoing topic of discussion. Given the global change toward sugar reduction in dietary guidelines, natural sweeteners such as monk fruit are becoming increasingly explored as prospective replacements. Organizations such as the WHO and FDA are recommending reduced intake of added sugar to reduce the risk of obesity, diabetes and cardiovascular disease [1,3]. The European Food Safety Authority (EFSA) is continuing to evaluate the safety of monk fruit as part of its Novel Foods program, which mandates that new food additives be evaluated rigorously before they are introduced to the market [3].

Despite its potential benefits, current research on MFE remains limited, particularly in large-scale human trials assessing long-term metabolic and cardiovascular effects. Unlike artificial sweeteners, monk fruit contains bioactive compounds that may exert health benefits beyond mere calorie reduction, but these require further validation through longitudinal clinical studies [3,13]. The sustainable application of monk fruit in food systems also depends on agricultural practices, supply chain stability, and consumer acceptance, which differ globally.

Future research should establish more precise guidelines on MFE’s metabolic impact, its role in glycemic control, and its safety profile over prolonged consumption. Moreover, comparative analyses between MFE and other sweeteners (both artificial and natural) should be conducted to guide public health policies and regulatory approvals. Policymakers should consider integrating monk fruit into dietary guidelines and public health programs aimed at reducing sugar consumption globally.

Although MFE is approved for use in US, China, and Canada, it has not yet been authorized in the EU. This regulatory barrier poses a significant challenge for international food manufacturers and developers looking to introduce MFE-sweetened products into the European market. As a result, innovation in sugar-reduced foods is slowed down, and consumer access to natural alternatives is limited. The lack of alignment between global regulatory regimes highlights the importance of coordinated safety assessments to support the broader implementation of MFE in health-focused food policies and product development.

#### 3.2.1. Metabolic and Glycemic Control

Several clinical trials have shown that MFE has a positive effect on glucose homeostasis and insulin response. A study by Tey et al. (2017) showed that MFE consumption led to an 18% reduction in glucose AUC and a 22% reduction in insulin AUC compared to sucrose (*p* < 0.05) [6]. Similarly, Epstein et al. (2024) showed that MFE consumption was linked to significant enhancement in glucose variability and insulin sensitivity, suggesting its potential in diabetes management and glycemic control [10].

#### 3.2.2. Symptom-Relief and Antioxidant Properties

Monk fruit extract has been shown to reduce markers of inflammation, contributing to better immune function and metabolic health. Elevated levels of IL-6 and TNF-α are associated with chronic low-grade inflammation, contributing to insulin resistance, impaired glucose metabolism and increased cardiometabolic risk. Therefore, the observed reductions in these markers after MFE consumption may reflect broader immunometabolic benefits beyond glycemic control. Wu et al. (2024) reported that MFE supplementation led to a 25% reduction in these inflammatory cytokines compared to placebo (*p* = 0.03), emphasizing its potential in mitigating chronic inflammation [12]. In addition, mogrosides in monk fruit show strongly antioxidant properties, which neutralize free radicals that promote oxidative stress and cell damage [11].

#### 3.2.3. Sustainable Health Implications

Monk fruit extract represents a sustainable alternative to artificial and caloric sweeteners, aligning with global health initiatives to reduce sugar consumption and combat metabolic diseases. Additionally, its cultivation requires a lower environmental input than traditional sugar production, making it a viable solution for sustainable food technology.

## 4. Discussion

This systematic review evaluates the health benefits, metabolic effects, and potential regulatory implications of monk fruit extract (MFE) based on randomized controlled trials (RCTs). While current evidence suggests that MFE has potential in glucose control, anti-inflammatory properties, and cardiovascular health, several limitations in the available studies must be considered.

Glucose and insulin responses differed between studies, which can be explained by the different study designs and crossover and parallel group formats, which may have influenced the results. The duration of the interventions also varied, with most studies being short term (lasting only a few days or weeks), which may not reflect long-term influence. Also, the dosage and form of MFE were varied (some used sweetened drinks, while others used lozenges or yoghurt), and the size or characteristics of the study populations varied as well. Some studies included only healthy individuals, while others included people with insulin resistance. A thorough assessment of the RCTs considered discloses discrepancies in methodology, sample size, duration of intervention, and study design that may impact the reliability and ability to generalize the results. A few key constraints for direct result comparison include:Heterogeneity in study design—The studies employed different trial designs, ranging from crossover to placebo-controlled trials, making direct comparisons challenging. While crossover designs help control inter-individual variability, they may not fully reflect long-term metabolic adaptations to MFE consumptionSmall trial size—The studies’ number of patients varied significantly (from n = 30 to n = 203), restricting the statistical power of individual studies to detect small but significant effects.Variation in MFE dosage and formulation—Some trials used pure monk fruit extract, while other studies incorporated MFE as part of botanical blends, making it hard to isolate its specific effects.Short study durations and the absence of long-term safety data—These represent a critical limitation across the included trials. Most interventions ranged from 7 to 15 days, which may be insufficient to assess sustained metabolic adaptations or detect delayed adverse effects. Consequently, the long-term impact of habitual MFE intake—particularly on glucose homeostasis, cardiovascular health, and gut microbiota—remains unclear and warrants further investigation.Population variability—While some studies included male and female participants, others were skewed toward one gender or specific metabolic profiles, limiting a broader applicability to general populations.Lack of ethnic diversity—Most trials were conducted in Asia and North America, making it uncertain whether findings can be generalized to European or other global populations with different dietary and genetic backgrounds.

The findings from this review align with previous research on non-nutritive sweeteners (NNSs), including steviol glycosides and erythritol, which have shown similar glucose-lowering effects and cardiometabolic benefits. Compared to artificial sweeteners like aspartame and sucralose, MFE appears to have a lower risk of metabolic disruption and does not contribute to insulin resistance [14].

Despite its demonstrated metabolic and health benefits, MFE remains unapproved in the European Union (EU) due to regulatory barriers outlined in the EU Novel Foods Regulation (EU 2015/2283) [3]. Unlike the U.S. GRAS status and China-approved food additive, the EU requires additional toxicological data before MFE can be legally marketed as a sweetener. The regulatory challenges faced by monk fruit resemble those previously encountered by steviol glycosides, which underwent years of regulatory scrutiny before approval in 2011. Greater alignment between global regulatory frameworks and industry-led applications will be crucial for integrating MFE into European markets.

MFE is widely used as a natural sweetener in several countries, particularly in the United States, Canada, China, and Australia, where it is available as an alternative to sugar and artificial sweeteners in beverages, baked goods, and processed foods. Given its high sweetness intensity (200–300 times sweeter than sucrose) and zero-calorie profile, MFE could play a key role in reducing added sugar intake and promoting healthier dietary patterns [13,14].

## 5. Conclusions

Our review shows that MFE may be a promising alternative to sugar and artificial sweeteners, offering metabolic benefits, anti-inflammatory properties, and potential cardioprotective effects but long-term clinical trials are needed to establish the safety profile of MFE and its potential role in chronic dietary interventions. The available RCTs demonstrate significant reductions in glucose levels, insulin resistance, and inflammatory markers, supporting their potential role in sustainable dietary strategies.

Moreover, monk fruit extract’s capacity to serve as a high-intensity sweetener with only minimal impact on blood glucose levels makes it an ideal candidate for innovation in the functional food and sugar reduction strategies. Its incorporation into food technology and diet plans can help support people managing diabetes, metabolic syndrome, and obesity, contributing to broader public health goals.

However, critical gaps remain, including the need for:Long-term safety data and larger sample sizes to assess the sustained effects of MFE on metabolic health;Regulatory approval in Europe;Expanding studies to diverse populations to improve generalizability;Addressing regulatory barriers through industry-driven safety evaluations;Investigating the gut microbiota impact of monk fruit extract compared to artificial sweeteners;Exploring its functional applications in beverages, baking, and processed foods as a natural sugar substitute.

By overcoming these challenges, monk fruit extract has the potential to become a key part of global health strategies aimed at curbing metabolic diseases while providing sustainability in food production. Its natural origin, metabolic benefits, and the potential for regulatory endorsement make it a valued tool for policymakers, researchers, and health practitioners concerned with functional food innovation.

## 6. Strength

This study is the first systematic review assessing the health effects of MFE in randomized controlled trials (RCTs). Following PRISMA guidelines, it includes high-quality trials evaluated using the Jadad Scale and Cochrane risk-of-bias assessment to ensure methodological rigor. The findings support sustainable health policies by emphasizing MFE as a natural, calorie-free substitute for sugar and artificial sweeteners. In addition, the study highlights the need for long-term studies with more heterogeneous populations to expand the evidence base for the metabolic and regulatory implications of MFE.

## Figures and Tables

**Figure 1 nutrients-17-01433-f001:**
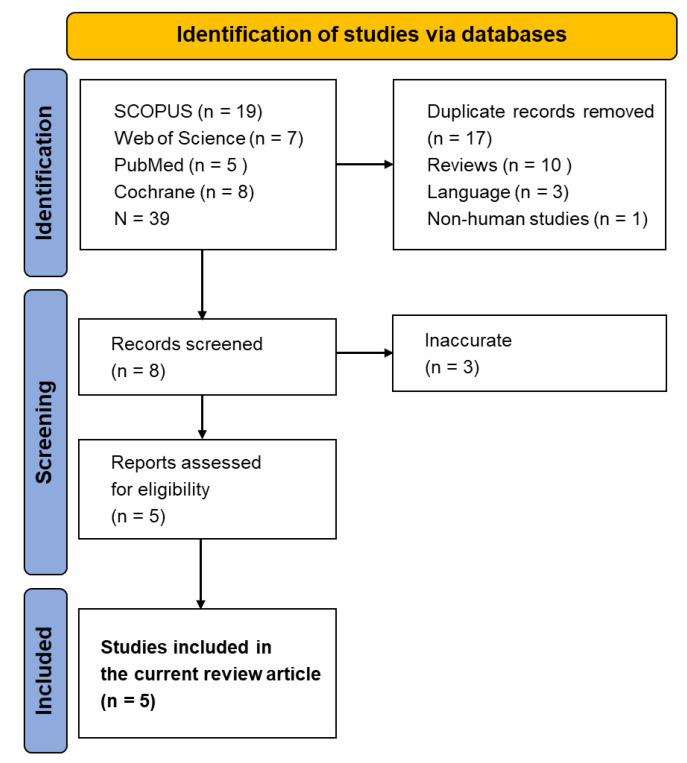
PRISMA 2020 flow diagram representing the screening strategy and selection process for eligible research articles.

**Table 1 nutrients-17-01433-t001:** Effects of monk fruit extract on glucose metabolism and insulin regulation.

**Factor**	**Country** **(Year)**		
	**Singapore** **(2017)**	**Singapore** **(2017)**	**USA** **(2024)**
Legal Considerations	Calorie-matched, FDA-approved	FDA-approved non-nutritive sweeteners	FDA GRAS-approved sweetener
Population	30 healthy adults M:F = 16:14; non-diabetic	30 healthy adults M:F = 16:14	50 adults M:F = 28:22; insulin-sensitive and insulin-resistant
Study Design	RCT, crossover	RCT, crossover	RCT, double-blind, crossover
Intervention	MFE, stevia, aspartame, sucrose beverages	MFE, stevia, aspartame, sucrose-sweetened beverages	MFE-sweetened yogurt
Duration	Single session	Single session	Single session
Outcome Measures	Postprandial glucose; insulin; energy intake; satiety response	Postprandial glucose, insulin AUC, energy intake	Reinforcing value of sugar; glucose response; insulin sensitivity
Key Findings	MFE ↓ glucose AUC −18%; ↓ insulin AUC −22% vs. sucrose	MFE ↓ glucose AUC −18%; ↓ insulin AUC −22% vs. sucrose	MFE ↓ sugar cravings −23%; ↓ fasting glucose −6%
Cochrane Risk of Bias	Low	Low	Low
Jadad Score	5	5	5
References	[6]	[9]	[10]

Jadad scale—0–5; M—male; F—female; AUC—area under the curve; ↓—decrease.

**Table 2 nutrients-17-01433-t002:** Inflammation-modulating potential and Symptom-Relief Effects of Monk Fruit Extract.

Factor	Country (Year)	
	China (2019)	China (2024)
Legal Considerations	EFSA-compliant herbal–medicinal use	Legally compliant for food testing
Population	203 patients (102 experimental, 101 control),	103 patients with chronic pharyngitis (52 experimental, 51 control)
Study Design	RCT, placebo-controlled	RCT, placebo-controlled
Intervention	Luo Han Guo decoction (experimental) vs. Black tea (control)	Botanical lozenge containing *Siraitia grosvenorii*,* Lonicera japonica*,* Platycodon grandiflorus*, *Glycyrrhiza uralensis*
Duration	48 h	15 days
Outcome Measures	Throat pain, hoarseness, mucosal congestion, cough, sputum production	Pharyngeal symptoms, adherence to treatment
Key Findings	Significant reduction in throat pain, hoarseness, and inflammation at 12, 24, and 48 h (*p* < 0.05). Cough and expectoration lower in the experimental group at 48 h (*p* = 0.003).	Significant reduction in pharyngeal symptoms (*p* < 0.01), improved congestion of pharyngeal mucosa (*p* < 0.05), higher treatment adherence in the experimental group
Cochrane Risk of Bias	Low	Moderate
Jadad Score	4	3
References	[11]	[12]

Jadad scale: 0–5.

## Data Availability

The data presented in this study are available at https://doi.org/10.18150/KHW5Q0.

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
