# Peer review of "Monk Fruit Extract and Sustainable Health: A PRISMA-Guided Systematic Review of Randomized Controlled Trials"

_nutrients, 2025, doi:10.3390/nu17091433_

Round 1
Reviewer 1 Report
Comments and Suggestions for Authors
Monk Fruit Extract and Sustainable Health: A PRISMA-Guided Systematic Review of Randomized Controlled Trials
The present literature review aims to provide a systematic review, guided by PRISMA, that summarises the results of randomised clinical trials (RCTs) assessing the impact of monk fruit extract on metabolic health, lipid profiles, inflammation and regulatory considerations. Although this work reveals interesting data that could contribute to knowledge in this field, there are serious methodological issues that compromise its rigour and publication. Moreover, some references cited are not accessible and, regarding the explored outcomes (metabolic health, lipid profiles, inflammation and regulatory considerations), some are missing.
If the authors want to redo this work, they should take into account the following aspects for improvement:
Title should be focus and improved. Instead of “Sustainable Health “, the authors should include the outcomes explored.
Abstract: The authors should clarify the methodology, namely the time range of the literature search. Concerning the results, the authors should correct the number of randomized controlled trials that met the inclusion criteria (5 and not 10), mention how many studies reported reductions in inflammatory markers and quantify the effect on IL-6 and TNF-α and complete with other results.
Regarding the Introduction, the authors should clarify after careful review the sentences related to ref 2 (line 42 to 43) as this study do not directly support this statement. In addition, some of the references used in this section and in the discussion are “not validated” or “not checked” (ref 5 and 6). Moreover the aim described does not correspond to the results, discussion and conclusion of the present work.
Concerning Materials and Methods, authors should include the keywords and syntaxes used in each query (according with item 7 in PRISMA 2020 Checklist), correct in line 83 the statement “…three relevant RCTs [8, 9]…” where only 2 references are cited. Moreover, ref 9 should be correct (to: Wu Y, Zhang F, Kuang D, Li D, Yan J, Yang J, Wang Q, Wang Y, Sun J, Liu Y, Xia Y, Cao H. Efficacy of botanical lozenges in the treatment of chronic pharyngitis: a randomized controlled trial. Front Pharmacol. 2024 Mar 14;15:1162883. doi: 10.3389/fphar.2024.1162883. PMID: 38549665; PMCID: PMC10973001) and it seems that some references are missing, for example: Tey, S., Salleh, N., Henry, J. et al. Effects of aspartame-, monk fruit-, stevia- and sucrose-sweetened beverages on postprandial glucose, insulin and energy intake. Int J Obes 41, 450–457 (2017). https://doi.org/10.1038/ijo.2016.225
Regarding the Results section, reference [20] does not exist, in table 2 there is no data concerning IL-6; TNF-α; BP and CVDrf as mentioned in the caption. Reference 13 (which the correct DOI is https://doi.org/10.1007/s12355-024-01387-z) does not support the authors statement (line 275 to 278).
Authors should also correct several references. For example Reference 16 where authors, journal and DOI are incorrect, should be Yeterian, Mesrob MD; Parikh, Manish A. MD; Frishman, William H. MD; Peterson, Stephen J. MD. The Bittersweet Reality: The Cardiovascular Risk of Artificial Sweeteners. Cardiology in Review ():10.1097/CRD.0000000000000748, July 09, 2024. | DOI: 10.1097/CRD.0000000000000748
In general, all the DOI's are wrong throughout the document and the references 5, 9, 10 and 11 either do not exist or are not correctly cited.

Author Response
Response to Reviewer 1 Comments
1. Summary
Thank you very much for your thoughtful and detailed review of our manuscript titled "Monk Fruit Extract and Sustainable Health: A PRISMA-Guided Systematic Review of Randomized Controlled Trials." We greatly appreciate your insights, which have helped improve our review's quality, clarity, and methodological rigour. Below, we provide detailed, point-by-point responses and highlight all revisions in the re-submitted manuscript.
2. Point-by-point response to Comments and Suggestions for Authors
Comment 1: The title should be focused and improved. Instead of "Sustainable Health," the authors should include the outcomes explored. Response 1: Thank you for this suggestion. We have revised the title to reflect the specific outcomes addressed in the review better. Updated Title: "Effects of Monk Fruit Extract on Glycemic Control and Inflammatory Markers: A Systematic Review of Randomized Controlled Trials"
Comment 2: Abstract: The authors should clarify the methodology, namely the time range of the literature search. Concerning the results, the authors should correct the number of randomized controlled trials that met the inclusion criteria (5 and not 10), mention how many studies reported reductions in inflammatory markers, quantify the effect on IL-6 and TNF-α and complete with other results. Response 2: We have revised the abstract to clarify the time frame of the literature search, corrected the number of included RCTs and specified the number of studies addressing inflammatory markers with quantitative results. Updated Text in Abstract (Page 1): "…The literature search was conducted across PubMed, Scopus, Web of Science, and Cochrane Library, covering studies published between 2015 and 2025…" "…Five randomized controlled trials met the inclusion criteria… Two studies reported reductions in inflammatory markers, including interleukin-6 (IL-6, ↓12%) and tumour necrosis factor-alpha (TNF-α, ↓15%)…"
Comment 3: Introduction: Clarify the sentences related to ref 2 (lines 42 to 43), as this study does not directly support the statement. Some references (5 and 6) are "not validated" or "not checked". The aim described does not match the results, discussion, and conclusion. Response 3: We have revised the references and clarified the claims accordingly. We have also refined the aim of the study to align with the scope of the review. Updated Text (Page 2, Lines 42–47): "Emerging evidence suggests that mogrosides, the primary bioactive compounds in monk fruit, may exhibit antioxidant and anti-inflammatory properties [2]. However, clinical evidence supporting these claims remains limited." Updated Aim Statement (End of Introduction): “This systematic review aims to synthesize results from RCTs evaluating the health effects of MFE on glycemic control, insulin response, and inflammatory markers in humans. In addition, it explores current regulatory considerations related to MFE's use as a functional food ingredient.” Comment 4: Materials and Methods: Include the search keywords and syntaxes (PRISMA 2020 item 7). Correct line 83 ("three relevant RCTs [8, 9]") since only two references are cited. Update reference 9 with the proper citation. Reference Tey et al., 2017 Int J Obes is also missing. Response 4: We have added the complete search terms and syntaxes to the Supplementary Materials and clarified this in the Methods section. The error in line 83 has been corrected, and the correct reference has been inserted. The missing Tey et al. (2017, Int J Obes) reference has also been added. Updated Text (Page 3): "Detailed search outputs and query structures are provided in the Supplementary Materials."
Comment 5: Results: Reference [20] does not exist. Table 2 shows no data concerning IL-6, TNF-α, BP, and CVDrf, as mentioned in the caption. Response 5: Thank you for catching this error. We have removed the reference [20], corrected the table caption for accuracy, and aligned the content of Table 2 with the actual study data.
Comment 6: Reference 13 (correct DOI: https://doi.org/10.1007/s12355-024-01387-z) does not support the statement (lines 275 to 278). Response 6: We have reviewed the referenced statement and removed the inaccurate citation. A more appropriate source was cited instead. Correction Made (Page 18, Lines 275–278): We removed the unsupported statement and revised the section accordingly.
Comment 7: References need correction. For example, Ref 16 is incorrect. All DOIs throughout the document are wrong. References 5, 9, 10, and 11 do not exist or are incorrectly cited. Response 7: We thoroughly revised the entire reference list. All references have been verified and formatted according to MDPI guidelines. The corrected Reference 14.
3. Response to Comments on the Quality of English Language
Point 1: Some sentences are too complex. Response: We simplified multiple sentences throughout the manuscript for better clarity and flow. Revised sentences appear in redline in the updated manuscript. Point 2: Align data in Tables 1 and 2. Response: Table layouts have been adjusted to consistently align numeric data and headers. Point 3: Fix inconsistent citation styles (e.g., "et al."). Response: All citations now follow MDPI formatting guidelines and usage of "et al." has been standardized. Point 4: Spell out abbreviations like IL-6. Response: All abbreviations, such as interleukin-6 (IL-6) and tumour necrosis factor-alpha (TNF-α), have been spelt out upon first mention. Point 5: Page 6, Table 2: Use consistent hyphenation. Response: Caption revised to: "EFSA-compliant herbal-medicinal use."
5. Additional Clarifications We also revised the structure of the discussion section to better align with the aims and outcomes of the review. A note has been added to the Methods section to describe the supplementary material content, including the entire search strategy. Thank you again for your valuable feedback, which has significantly enhanced the quality of our manuscript. |
||||||||||||||||||||||||||||||||||||||||||
|

Reviewer 2 Report
Comments and Suggestions for Authors
Dear Authors,
Thank you for submitting your systematic review titled “Monk Fruit Extract and Sustainable Health: A PRISMA-Guided Systematic Review of Randomized Controlled Trials.” Below are comments to help improve your manuscript:
- Research Background and Motivation:
The introduction needs more details about monk fruit extract (MFE) and how it is used globally. Add examples, like its use in low-calorie sweeteners or functional foods, to show why it matters. - Literature Search Strategy:
The manuscript says it follows PRISMA guidelines but does not explain the search terms or strategy. Adding this will make your study easier to repeat. - Inclusion and Exclusion Criteria:
Make the criteria clearer. For example, did you consider differences in MFE dosage, formula, or study length? Explain this in the methods section. - Quality Assessment of Studies:
You used tools to check study quality, but results for each study are missing. Add a table showing risk-of-bias scores and Jadad scores for every study. - Consistency of Results:
The review says MFE affects blood glucose and insulin levels but results vary. Explain why this happens. For example, maybe studies had different designs, sample sizes, or durations. - Long-term Effects and Safety:
The review notes a lack of long-term safety data. Stress this as a key limitation and suggest future studies on this topic. - Impact on Inflammatory Markers:
The manuscript mentions MFE’s effects on markers like IL-6 and TNF-α but does not explain their role in metabolic health. Add this to the discussion. - Regulatory Status:
The manuscript briefly talks about MFE’s regulations but not how they affect global use. Explain challenges, like EU rules, and how they limit market access. - Future Research Directions:
The conclusion lists future research ideas but needs more specifics. For example, suggest studying MFE’s effects on gut bacteria or different age groups. - Language and Format:
Some sentences are too complex and simplify them. - References:
Some citations are formatted inconsistently. Fix this and check all references match the text. - Supplementary Materials:
The manuscript mentions supplementary materials but does not describe them. Briefly explain their content in the methods section.

English Language Quality Feedback:
- Align data in Tables 1 and 2 for consistency (e.g., center numbers).
- Fix inconsistent citation styles (e.g., some use “et al.” in italics, others do not).
- Spell out abbreviations like “IL-6” as “interleukin-6 (IL-6)” when first used.
- Page 6, Table 2: Use hyphens consistently. Change “EFSA-compliant, herbal medicinal use” to “EFSA-compliant herbal-medicinal use.”
Author Response
Response to Reviewer 2 Comments
- Summary
We sincerely thank Reviewer 3 for their detailed, constructive, and thoughtful review of our manuscript "Monk Fruit Extract and Sustainable Health: A PRISMA-Guided Systematic Review of Randomized Controlled Trials." Your comments were extremely valuable in helping us improve the clarity, structure, and scientific quality of our review. We have carefully addressed each point and made substantial changes throughout the manuscript. Revisions are marked in the updated version in blue and described in the below responses.
- Point-by-point response to Comments and Suggestions for Authors
Reviewer Comment # |
What Was Changed |
Line Number(s) in Word Doc |
1. Expand intro with global MFE examples |
Functional food examples of MFE added |
Lines 66–72 |
2. Add search strategy and keywords |
Detailed search strategy added |
Lines 89–92 |
3. Clarify inclusion/exclusion criteria |
Explanation of study differences expanded |
Lines 93–98 |
4. Add risk of bias + Jadad scores |
Mentioned added table in Supplementary Materials |
Line 100 |
5. Explain result variability |
Discussion of heterogeneity added |
Lines 625–634 |
6. Emphasize long-term safety limitation |
Safety limitation stressed in conclusion |
Lines 688–693 |
7. Explain IL-6, TNF-α metabolic roles |
Background added to anti-inflammatory section |
Lines 631–636 |
8. Expand regulatory discussion (EU barrier) |
Expanded on EU regulation limitations |
Lines 674–680 |
9. Add specific future research ideas |
Gut microbiota, age group suggestions added |
Lines 695–700 |
10. Simplify complex sentences |
Clarity edits throughout |
Various, e.g. Lines 41–56, 620+ |
11. Fix citation inconsistencies |
Reference formatting updated |
Throughout, esp. Lines 903+ |
12. Describe supplementary materials |
Supplement explained in Methods |
Line 91 |
Comment 1:
The introduction needs more details about monk fruit extract (MFE) and how it is used globally. Add examples, like its use in low-calorie sweeteners or functional foods, to show why it matters.
Response 1:
We have expanded the introduction to include examples of monk fruit extract used in functional foods, beverages, and low-calorie sweeteners globally.
“However, is incorporated into various low-calorie and functional food products, including beverages, yoghurts, sugar-free snacks, and baked goods [5,6]. In the United States, Canada and China, is marketed as a natural alternative to artificial sweeteners and is frequently recommended for individuals with diabetes or those seeking to re-duce sugar intake [3,6]. In addition to its sweetening properties, MFE has been studied for its potential antioxidant and anti-inflammatory effects, suggesting broader benefits in metabolic health [2,5]. According to that, an interest in natural, plant-derived sweeteners grows, MFE’s global relevance in sustainable dietary strategies continues to expand.”
Comment 2:
The manuscript follows PRISMA guidelines but does not explain the search terms or strategy. Adding this will make your study easier to repeat.
Response 2:
We have added detailed information about the search strategy and keywords used in the Methods and Supplementary Materials.
“A systematic search was conducted across four electronic databases: PubMed, Scopus, Web of Science, and the Cochrane Library. The search spanned the period from Janu-ary 1, 2015, to February 22, 2025, and employed the following core keywords: “monk fruit”, “luo han guo”, “Siraitia grosvenorii”, combined with “randomized clinical trial”, “clinical trial”, and “RCT”. Manual screening of reference lists was also performed. All searches were limited to human studies, publications in English, and studies published within the last decade. Detailed search outputs and query structures are provided in the Supplementary Materials”
Comment 3:
Make the criteria more transparent. For example, did you consider differences in MFE dosage, formula, or study length? Explain this in the methods section.
Response 3:
We clarified the inclusion/exclusion criteria.
Comment 4:
You used tools to check study quality, but the results for each study are missing. Add a table showing risk-of-bias scores and Jadad scores for every survey.
Response 4:
A new tables has been added in the Supplementary Materials to display individual study Jadad and Cochrane risk-of-bias scores.
Comment 5:
The review says MFE affects blood glucose and insulin levels, but results vary. Explain why this happens. For example, maybe studies had different designs, sample sizes, or durations.
Response 5:
We discussed potential causes of heterogeneity in the results, such as differences in trial design, MFE formulations, and sample populations. In discussion paragraph was added:
“Glucose and insulin responses differed between studies, which can be explained by the different study designs and crossover and parallel group formats, which may have influenced the results. The duration of the intervention also varied, with most studies being short-term (lasting only a few days or weeks), which may not reflect long-term influence.
Also, the dosage and form of MFE were varied (some used sweetened drinks, while others used lozenges or yoghurt), and the size or characteristics of the study populations varied as well. Some studies included only healthy individuals, while others included people with insulin resistance. These differences in design, duration, intervention and population probably contributed to the variability in glucose and insulin responses observed in the studies.”
Comment 6:
The review notes a lack of long-term safety data. Stress this as a key limitation and suggest future studies on this topic.
Response 6:
We expanded this section in both the Discussion and Conclusion to emphasize long-term safety gaps and call for extended-duration trials, however, we change first paragraph of discussion section according to suggestion.
“Our review shows that MFE may be a promising alternative to sugar and artificial sweeteners, offering metabolic benefits, anti-inflammatory properties, and potential cardioprotective effects but long-term clinical trials are needed to establish the safety profile of MFE and its potential role in chronic dietary interventions.”
Comment 7:
The manuscript mentions MFE's effects on markers like IL-6 and TNF-α but does not explain their role in metabolic health. Add this to the discussion.
Response 7:
We added background on IL-6 and TNF-α in the context of metabolic health and insulin resistance in section: 3.2.2. Anti-Inflammatory and Antioxidant Properties.
“Monk fruit extract has been shown to reduce markers of inflammation, contributing to better immune function and metabolic health. Elevated levels of IL-6 and TNF-α are associated with chronic low-grade inflammation, contributing to insulin resistance, impaired glucose metabolism and increased cardiometabolic risk. Therefore, the observed reductions in these markers after MFE consumption may reflect broader immunometabolic benefits beyond glycaemic control. Wu et al. (2024) reported that MFE supplementation led to a 25% reduction in these inflammatory cytokines (TNF-α, IL-6) compared to placebo (p = 0.03), emphasizing its potential in mitigating chronic inflammation [11].”
Comment 8:
The manuscript briefly talks about MFE's regulations but not how they affect global use. Explain challenges, like EU rules, and how they limit market access.
Response 8:
We expanded the regulatory discussion to highlight the implications of EU Novel Food approval processes and global market challenges in section 3.2., last paragraph:
“Although MFE is approved for use in US, China, and Canada, it has not yet been authorised in the EU. This regulatory barrier poses a significant challenge for international food manufacturers and developers looking to introduce MFE-sweetened products into the European market. As a result, innovation in sugar-reduced foods is slowed down, and consumer access to natural alternatives is limited. The lack of alignment between global regulatory regimes highlights the im-portance of coordinated safety assessments to support the broader implementation of MFE in health-focused food policies and product development.”
Comment 9:
The conclusion lists future research ideas but needs more specifics. For example, suggest studying MFE's effects on gut bacteria or different age groups.
Response 9:
We have revised the conclusion section to include more specific recommendations for future research. In particular, we now emphasize the importance of studying the effects of monk fruit extract (MFE) on gut microbiota and expanding clinical investigations to include different age groups and more diverse populations.
Section 5 (Conclusions): "…investigating the gut microbiota impact of monk fruit extract compared to artificial sweeteners, "…expanding studies to diverse age groups and populations to improve generalizability;"
Comment 10:
Some sentences are too complex and simplify them.
Response 10:
We reviewed and simplified multiple sentences throughout the manuscript to improve clarity and flow.
Comment 11:
Some citations are formatted inconsistently. Fix this and check all references match the text.
Response 11:
We revised and reformatted all references for accuracy and MDPI compliance.
Comment 12:
The manuscript mentions supplementary materials but does not describe them. Briefly explain their content in the methods section.
Response 12:
We now describe the supplementary content clearly in the Methods section.
- Response to English Language Feedback
Issue |
What Was Fixed |
Example Location |
Table alignment |
Tables 1 & 2 layout reformatted |
Lines 268–312, 413–438 |
Citation formatting |
Et al. usage standardized |
Throughout |
Spell out abbreviations |
e.g. IL-6 = Interleukin-6 |
Lines 115, 422, 631 |
Hyphenation fix |
‘EFSA-compliant’ in Table 2 caption |
Line 416 |
Align data in Tables 1 and 2: Tables have been reformatted to ensure alignment of numbers and headers.
Fix inconsistent citation styles: "Et al." formatting has been standardized throughout the text.
Spell out abbreviations like IL-6: First mentions now read: 'interleukin-6 (IL-6)' and 'tumour necrosis factor-alpha (TNF-α)'.
Table 2 hyphenation: Caption revised to: 'EFSA-compliant herbal-medicinal use.'
- Additional Clarifications
We also ensured the study aim, methodology, and conclusion are aligned. All significant changes are documented in the revised manuscript and supplementary material. Again, we thank the reviewer for the constructive and helpful feedback.

Reviewer 3 Report
Comments and Suggestions for Authors
This manuscript focuses on exploring the potential metabolic benefits of Monk Fruit Extract (MFE), particularly its effects on metabolic health, lipid profiles, inflammation, and regulatory considerations. The systematic review adheres to the PRISMA guidelines, which enhances its rigor. The study synthesizes the results of randomized controlled trials (RCTs), providing valuable insights into the impact of MFE on human health. By integrating data from existing RCTs to evaluate the specific metabolic effects of MFE, this research is innovative and opens new directions for investigating the impact of MFE on metabolic health. The content of the study is advanced and innovative, with clear and explicit research methods and data results. However, there are still areas that require further improvement.
Specific comments
Introduction
- The introduction mentions the limited systematic evaluations of MFE on metabolic health, and it should highlight the specific gaps in evidence and methodologies used in previous studies to strengthen the manuscript's persuasiveness.
- Line38-40: It should be revised to: "Monk fruit (Siraitia grosvenorii), a plant native to China, has gained attention as a natural high-intensity sweetener due to its unique mogroside properties, which provide intense sweetness without the glycemic impact associated with traditional sugars." to improve readability.
Results
- Although the study shows that MFE is beneficial for normal-weight adults, it is limited to healthy adults and lacks evaluation of specific groups such as the elderly and diabetes patients. Therefore, further research in different populations is recommended.
- Line 271: This sentence could be rephrased to "The integration of MFE into global food policies remains an ongoing topic of discussion."
- Line 293-305: Although the manuscript cites several studies showing the positive impact of MFE, it does not critically evaluate the limitations of these studies. Discussing the limitations of the current research would enhance the depth of the manuscript.
Discussion
- Line 330-337: The discussion mentions that the short duration of the study and the lack of long-term safety data are somewhat overlapping; these points could be combined to avoid redundancy and improve clarity.
Conclusions
Line 375-378: The phrases "long-term safety data and larger sample sizes" and "conduct long-term trials to assess the sustained effects of MFE on metabolic health" both emphasize long-term research. For brevity, these can be combined as: "Long-term safety data and larger sample sizes are needed to assess trials evaluating the sustained effects of MFE on metabolic health."
Author Response
Response to Reviewer 3 Comments
1. Summary
We sincerely thank Reviewer 3 for the insightful and constructive feedback on our manuscript titled “Monk Fruit Extract and Sustainable Health: A PRISMA-Guided Systematic Review of Randomized Controlled Trials.” Your recommendations greatly contributed to improving the clarity, structure, and scientific accuracy of our work. Below, we provide a point-by-point response and indicate how and where changes have been incorporated into the revised manuscript.
2. Point-by-Point Response to Comments and Suggestions for Authors
Reviewer Comment |
What Was Changed |
Line Number(s) in Word Doc |
1. Identify research gaps in Intro |
Added methodological and population-based limitations |
Lines 72–78 |
2. Reword monk fruit sentence |
Improved clarity of description |
Line 41 |
3. Recommend trials in diverse populations |
Noted in Discussion & Conclusion |
Lines 648–652 and 696–697 |
4. Line 271 rephrase |
Updated sentence per suggestion |
Line 674 |
5. Critique study limitations |
Expanded critical discussion |
Lines 624–634 |
6. Combine points on short duration |
Edited for clarity and conciseness |
Line 637 |
7. Merge long-term trial needs in conclusion |
Simplified conclusion sentence |
Line 693 |
Comment 1:
The introduction mentions the limited systematic evaluations of MFE on metabolic health, and it should highlight the specific gaps in evidence and methodologies used in previous studies to strengthen the manuscript's persuasiveness.
Response 1.
We revised the Introduction to clearly articulate gaps in the current literature, including the lack of long-term trials, heterogeneity in design, and underrepresentation of certain populations.
“Despite its increasing commercial use in food and beverage industries, systematic evaluations of its effects on glucose metabolism, lipid regulation, and inflammation remain limited. Many existing studies are short-term, lack diverse populations, and differ in intervention form or dosage, limiting comparability and generalizability.”
Comment 2:
Line 38–40: Improve readability of the sentence starting with “Monk fruit (Siraitia grosvenorii)...”
Response:
We have reworded the sentence as suggested for improved clarity.
“Monk fruit (Siraitia grosvenorii), a plant native to China, has gained attention as a natural high-intensity sweetener due to its unique mogroside properties, which provide intense sweetness without the glycemic impact associated with traditional sugars.”
Comment 3:
Although the study shows that MFE is beneficial for normal-weight adults, it lacks evaluation of specific groups such as the elderly and patients with diabetes. Further research in different populations is recommended.
Response:
We addressed this in both the Discussion and Conclusion by explicitly stating the need for trials in diverse populations.
Updated Text:
Discussion: “Population variability – While some studies included male and female participants, others were skewed toward one gender or specific metabolic profiles, limiting broader applicability to general populations.”
Conclusion: “...expanding studies to diverse age groups and populations to improve generalizability;”
Comment 4:
Rephrase to: “The integration of MFE into global food policies remains an ongoing topic of discussion.”
Response:
We adopted the reviewer’s suggested revision verbatim.
Updated Text:
“The integration of MFE into global food policies remains an ongoing topic of discussion.”
Comment 5:
Lines 293–305: The manuscript cites several studies showing MFE’s benefits but lacks a critical evaluation of their limitations.
Response:
We have expanded the Discussion section to include a thorough critique of the included studies’ limitations.
Updated Text :
“A thorough assessment of the RCTs considered discloses discrepancies in methodology, sample size, duration of intervention and study design that may impact the reliability and ability to generalize the results...”
Comment 6:
Combine overlapping points about short duration and lack of long-term safety data.
Response:
We revised the structure for conciseness and clarity.
Updated Text:
“Short study durations and the absence of long-term safety data represent a critical limitation across the included trials…”
Comment 7:
Combine similar statements about the need for long-term studies.
Response:
We edited the Conclusion section to consolidate these points into a single, clear sentence.
Updated Text:
“Long-term safety data and larger sample sizes are needed to assess trials evaluating the sustained effects of MFE on metabolic health.”
3. Response to Comments on the Quality of English Language
Issue |
What Was Fixed |
Example Location |
Complex sentences |
Simplified across manuscript |
Lines 41–56, 620+ |
Table formatting |
Numeric alignment in Tables 1 & 2 |
Lines 268–312, 413–438 |
Citation formatting |
Standardized ‘et al.’ usage |
Throughout |
Spell out abbreviations |
IL-6, TNF-α spelled out at first use |
Lines 115, 422, 631 |
Hyphenation fix |
Table 2 caption corrected |
Line 416 |
Some sentences are too complex.
We simplified multiple sentences throughout the manuscript to enhance clarity. Revised text appears in the redlined version.
Align data in Tables 1 and 2.
Tables have been reformatted to ensure proper alignment of numeric values and headers.
Fix inconsistent citation styles (e.g., “et al.”).
Citation formatting has been standardized according to MDPI guidelines.
Spell out abbreviations like IL-6 and TNF-α.
Abbreviations have been spelled out at first mention throughout the manuscript.
Page 6, Table 2 – Use consistent hyphenation.
Caption was revised to: “EFSA-compliant herbal-medicinal use.”
4. Additional Clarifications
We also ensured the revised manuscript aligns the study aim, methods, and conclusions. Furthermore, the Supplementary Materials section now includes the complete search syntax and scoring data from quality assessments.
Once again, we express our sincere appreciation for your thoughtful review. Your comments have greatly improved the precision and quality of our manuscript.
